# Treasure on the Earth—Gold Nanoparticles and Their Biomedical Applications

**DOI:** 10.3390/ma15093355

**Published:** 2022-05-07

**Authors:** Justyna Milan, Klaudia Niemczyk, Małgorzata Kus-Liśkiewicz

**Affiliations:** Institute of Biology and Biotechnology, College of Natural Sciences, University of Rzeszow, St. Pigonia 1, 35-310 Rzeszow, Poland; justii.milan@gmail.com (J.M.); klaudia.niemczyk1@gmail.com (K.N.)

**Keywords:** diagnosing, gold nanoparticles, imaging, photodynamic therapy, photothermal therapy

## Abstract

Recent advances in the synthesis of metal nanoparticles (NPs) have led to tremendous expansion of their potential applications in different fields, ranging from healthcare research to microelectronics and food packaging. Among the approaches for exploiting nanotechnology in medicine, gold nanomaterials in particular have been found as the most promising due to their unique advantages, such as in sensing, image enhancement, and as delivery agents. Although, the first scientific article on gold nanoparticles was presented in 1857 by Faraday, during the last few years, the progress in manufacturing these nanomaterials has taken an enormous step forward. Due to the nanoscale counterparts of gold, which exhibit distinct properties and functionality compared to bulk material, gold nanoparticles stand out, in particular, in therapy, imaging, detection, diagnostics, and precise drug delivery. This review summarizes the current state-of-the-art knowledge in terms of biomedical applications of gold nanoparticles. The application of AuNPs in the following aspects are discussed: (i) imaging and diagnosing of specific target; (ii) treatment and therapies using AuNPs; and (iii) drug delivery systems with gold nanomaterials as a carrier. Among the different approaches in medical imaging, here we either consider AuNPs as a contrast agent in computed tomography (CT), or as a particle used in optical imaging, instead of fluorophores. Moreover, their nontoxic feature, compared to the gadolinium-based contrast agents used in magnetic resonance imaging, are shown. The tunable size, shape, and functionality of gold nanoparticles make them great carriers for targeted delivery. Therefore, here, we summarize gold-based nanodrugs that are FDA approved. Finally, various approaches to treat the specific diseases using AuNPs are discussed, i.e., photothermal or photodynamic therapy, and immunotherapy.

## 1. Introduction

Nanotechnology is a multidisciplinary science that deals with materials science, physics, and chemistry, among others. Applications of nanotechnology cover most branches of science and technology. Depending on the component they are made of, their properties, morphology, or their size, nanomaterials can be divided into polymer nanomaterials, carbon nanomaterials, metal nanomaterials, lipid nanomaterials, and semiconductor nanomaterials [1]. Inorganic nanoparticles, the structures of which exhibit significantly different physical, chemical, and biological properties and functionality, as opposed to their corresponding mass counterparts, have attracted more and more interest. In particular, looking at noble metal nanoparticles, it has been noticed that their electromagnetic, optical, and catalytic properties are strongly influenced by shape and size. This has primarily led to an increase in research into their synthesis pathways, which will allow for better shape and size control in various nanotechnology applications [2].

Metal nanoparticles are increasingly being used in biomedical fields due to their small size-to-volume ratio, functionalization, stabilization, and ease of detection [3]. Among inorganic metal nanomaterials, gold nanoparticles stand out, the distinct physical and chemical properties of which make them ideal for various applications, especially in therapy, detection and diagnostics, or precise drug delivery [4]. There are various methods of obtaining metal nanoparticles. The synthesis of nanoparticles is divided into two basic strategies, “top-down” and “bottom-up” approaches. Top-down methods include synthesizing AuNP from a bulk material and breaking it down into smaller particles. The bottom-up approach involves the synthesis of nanoparticles starting from the level of molecules or atoms. The synthesis of nanoparticles is also divided into physical, chemical and biological methods [3,5]. The production of monodisperse and size-controlled gold nanoparticles and is of great importance. AuNPs are usually synthesized using a colloidal method, where, through the use of reducing agents, i.e., a metal precursor, gold ions are reduced to gold nanoparticles, and, in the presence of appropriate stabilizing agents, agglomeration of the particles is prevented and this allows easy adjustment of the size, shape, or even optical properties of the nanostructure [6,7]. Other approaches are green synthesis methods, where fungi or bacterial extracts (intracellular or extracellular), microorganisms, plant extracts, or biomolecules are used in the synthesis of AuNPs [8,9]. The biosynthesis of metal nanoparticles is gaining more importance due to its simple, rapid, easily-reproducible, environmentally friendly, ecological, and cost effective procedure [10].

The unique properties of nanoparticles make them promising candidates for a wide range of applications [11]. Gold nanoparticles (AuNPs) have the ability to absorb and scatter light, and can convert optical energy into heat using nonradiative electron relaxation dynamics and surface chemistry. Moreover, gold nanoparticles can be used as drug carriers, making them very attractive and versatile nanoparticles. The features of AuNPs that make them particularly attractive in biomedicine are their excellent stability and biocompatibility, ease to functionalize their surfaces, their low toxicity, and their drug transferability. Other features, such as shape and size adaptation, have certainly drawn attention for the use of gold nanoparticles in many fields. These nanoparticles are very small and comparable in size to many biomolecules [7]. Metal nanoparticles, the dimensions of which vary between 2 and 20 nm, are similar in size to the dimensions of biomolecules, such as antigens, antibodies, and other proteins or DNA [12]. The size of gold nanoparticles most often ranges from a few nanometers to about several hundred nanometers, along the narrowest two dimensions, and the shapes can be solid spheres, rods, chains, cages, stars, and shells. The shape and size of nanoparticles affect their physical characteristics and surfaces, such as the color of the NP, and how the body interacts with the nanoparticle [13].

With the growing interest in the use of gold nanoparticles in biomedicine, so too do concerns about human safety increase. It has become crucial to understand nano–bio interactions and the resulting toxicity of nanomaterials, and to understand the adverse effects of exposure to these novel materials [14]. Some of the most important factors influencing the toxicity of NPs are their size and route of administration (Figure 1) [11]. 

Research proves that the best way to introduce AuNPs is through injection into the bloodstream, not only because of the low toxic effect, but also because of the simplicity of this procedure. Zhang et al. additionally observed a significant decrease in body weight, spleen index, and red blood cells associated with oral administration of AuNPs [15]. Oral administration of nanoparticles is also associated with a low absorption. However, after intravenous administration, a low gold excretion is observed in feces or urine because nanoparticles are phagocytosed by Kupffer cells of the reticuloendothelial system, where they will remain for a long time [16,17]. Wang et al. reported on the in vivo synthesis of AuNPsafter administered into the mice. The materials were absorbed by the gastrointestinal tract and reached remote infected lesions. They exhibited antimicrobial properties and an effective clearance, as well as biocompatibility. These results showed that these nanoparticles do not adversely affect the functioning of the liver, kidneys, or immune system [18]. Kumari et al. tested the possibility of administering apple polysaccharide-modified gold nanoparticles in the treatment of type 1 diabetes mellitus. AuNPs-conjugated insulin (AuNPs-INS) at high doses caused a 3.36-fold decrease in blood glucose levels within 240 min, while oral administration of INS did not lower blood glucose levels. The 28-day study also showed a better improvement in body weight, lipid profile, urea, creatinine, and hepatic parameters with AuNP-INS, for which the observed value was similar to intraperitoneal insulin [19]. 

After getting into the bloodstream, nanoparticles are exposed to an environment rich in proteins, cells, and tissues. The endothelium of the lung and muscle capillaries allows for the transport of only small particles (<3 nm), while the endothelium of the kidneys, liver, and spleen have larger spaces, which increases the entry of nanoparticles with a hydrodynamic diameter below 60 nm [11]. Smaller NPs are also more likely to penetrate cells (Figure 2) and can be detected in cell nuclei, which translates into greater DNA damage [20]. Subcutaneous, intramuscular, or topical administration of colloidal drug carriers generally retains the drug carrier for a longer time than the free drug, and the drug carriers are primarily retained by local lymph nodes. In vivo distribution after administration depends heavily on the size and shape of the NP, the surface charge and the surface hydrophobicity [21]. 

Among the extensive amount of research, there are also contradictory results regarding the toxicity of gold NPs, depending on their size and shape. For instance, Li et al. reported higher toxicity of small NPs with sizes between 6 and 24 nm, compared to NPs with sizes 42–61 nm [22]. On the contrary, other studies have shown the opposite effect. Among AuNPs, ranging in size from 3 to 100 nm, smaller ones do not cause mortality of mice; mice treated with bigger ones suffered loss of appetite and weight, and, after 21 days, the mice died [23]. Enea et al. highlighted the important problem of using different units when comparing the toxicity of different sizes of gold nanoparticles. The authors postulated that concentration should be expressed in terms of number of AuNPs/volume unit instead of the most frequently used Au atom concentration (in mass or mol/volume unit) to avoid biased results [24]. For smaller-sized NPs, a higher amount of AuNPs is available for uptake and for inducing cytotoxicity, in spite of the same Au content. Gold nanoparticles can exist in many different forms, such as nanospheres, nanotriangles, nanoprisms, and nanorods [25]. Moreover, during synthesis, they can be modified with various molecules. Based on this, their toxicological profiles may be different. A systematic multiparametric comparative study was performed to assess the influences of size, shape, and coating agents of gold nanomaterials on the toxicokinetic and toxicodynamic profiles [24]. The experiments were performed on the human cerebral microvascular endothelial cell line (hCMEC/D3), an in vitro model of the human blood–brain barrier (BBB). Data showed that toxicity was greater for star-shaped relative to sphere-shaped AuNPs, and the citrate coating was more toxic than 11-mercaptoundecanoic acid. Functionalization of the AuNPs may have increased their potential for use as antimicrobial materials, which can be further immobilized onto medical devices. Indeed, Comune et al. assessed the antimicrobial properties and proangiogenic potential of soluble and immobilized LL37 peptides (conjugated to gold nanoparticles). Both soluble and immobilized peptides have antimicrobial activity; however, immobilized LL37 peptide showed less cytotoxicity toward endothelial cells (ECs). In addition, although both of the tested samples showed proangiogenic activities, they induced different signaling pathways. LL37-AuNPs also have superior wound healing properties compared to soluble peptide, both in vitro and in vivo systems [26,27]. 

## 2. Imaging and Diagnosing

Increasingly, refined metal (gold) nanoparticles are being applied in imaging to visualize key subcellular compartments and/or for diagnostics (Figure 3). The fundamental aim of imaging is to recognize and localize specific targets, very often by accumulation of a specific imaging compound at the specific part of the cell/body or disease site, in a harmless and non-invasive way [28]. For in vivo imaging, either high avidity of the NPs in target binding or their payload delivery and capacity for multiplexing with therapeutics and other agents are highly desired. Moreover, the ability of NPs to induce changes in electromagnetic or sound waves to transmit biological signals to monitoring devices is also desired [29]. Among the various metal NPs, owing to their high atomic number, high X-ray absorption coefficient, and unique optical properties, gold nanoparticles have received considerable attention. These make this nanomaterial ideal for electron microscopy, computed tomography, or colorimetric-biosensing approaches [30,31].

Computed tomography (CT), a non-invasive system, is founded on the exploitation of X-ray scanning, its attenuation in tissues, and computed image reconstruction to obtain morphologic and vascular information within the body [32,33]. This is obtained due to the fact that various tissues have different affinities to absorb X-rays [34,35]. The absorption and dispersion of X-ray radiation are, collectively, referred to as attenuation. The attenuation of any substance in CT imaging is defined on the Hounsfield scale and is expressed in terms of Hounsfield units according to the following equation: Attenuation (HU) = 1000 × (μ_x_ − μ_water_)/(μ_water_ − μ_air_) [36]. Currently, in CT imaging, contrast agents predominantly based on iodine containing molecules, are used to enhance absorbing X-rays. However, they are not specifically targeted since they cannot be bioconjugated with other components. Moreover, they exhibit very short imaging times due to rapid clearance by the kidney and poor contrast in large patients [37,38]. Thus, the combination of CT with gold nanoparticles as a contrast agent, due to the strong X-ray attenuation and bioconjugation, has recently seen significant development [39]. One of the key factors determining the effectiveness of CT contrast agents is a high atomic number—the higher it is, the better the resulting CT contrast. This makes AuNPs ideal candidates for CT contrast agents, as the high atomic number of gold (Z = 79) can induce strong X-ray suppression [39]. Galper et al. reported that gold nanoparticles provide almost 1.9 times greater contrast than iodine when scanning at 120 kV in water with a Brilliance iCT scanner with a gold NP attenuation of 5.14 HU/mM [40]. These results are consistent with other research where, for gold at a 120 kV attenuation rate, the value was 5.4 HU/mM, and for iodine it was 2.2 HU/mM [41]. For instance, the AuNP colloid has been introduced for the imaging of cardiovascular diseases. This is due to its extended blood circulation time, which, in turn, allows longer imaging times and the delineation of blood vessels [38]. The great advantage of gold nanoparticles is the significantly longer blood half-life. For commercially available iodine preparations, it is less than 10 min, while for AuNP-PEG it can reach 14.57 ± 3.27 h [42]. Nanosized gold can passively accumulate in tumor tissues more readily than in normal tissues due to the enhanced permeability and retention effect (EPR) of tumors. Therefore, an improvement in the contrast between normal and cancer cells can be observed [43,44]. Moreover, AuNPs can be actively targeted to a specific type of cancer by loading antibodies, peptides, or ligands on them, which enables tumor detection with CT imaging [39]. Usually, antibodies bound to conventional CT contrast agents (i.e., iodine) have not provided sufficient targeting CT contrast and load around three iodine atoms per antibody. Thus, using AuNPs seems to be much more promising since these nanoparticles can load antibodies with greater numbers of heavier atoms [45]. Gold nanoparticles are also consider as an ideal blood pool CT agents due to their ability to overcome biological barriers and remain in the blood stream for a prolonged period of time, compared to molecules containing iodine [45]. However, the incapability to uptake into the brain, due to the BBB, is the main problem in the treatment and diagnosing of neurodegenerative diseases. Shilo et al. demonstrated that, after injection of the insulin-targeted gold NPs into rat, they were found in a specific brain region using a micro-CT scanner. Therefore, gold NPs have been suggested for imaging and therapy of neurodegenerative disorders as a nanodevice that can overcome the restrictive mechanism of the BBB [46].

Optical imaging, as a much more rapid and inexpensive method, is one of the most preferable techniques since optical microscopy approaches can achieve astounding spatial resolutions [47]. This can be achieved through the generation of colorimetric contrast between specific cells/organelles/biocompounds by these imaging agents, mostly through the use of organic fluorophores [31]. However, the limits of the latter method are prone to photobleaching and a broad emission window [48]. Moreover, compared with typical fluorophores, gold nanoparticles show a light-scattering power equivalent to the signal arising from nearly 106 fluorescein molecules, and, more importantly, does not photobleach [49]. Thus, gold NPs, with their unique optical properties originating from surface plasmon resonance (SPR), are considered an excellent biomedical diagnostic tool due to their large absorption and scattering cross-sections when excited [50]. With the development of microscopic methods, gold nanoparticles have been extensively used for the direct or indirect visualization of biological systems. Starting from imaging the intracellular location of the AuNPs inside cells, through the real-time tracking of biointeractions, ending with measuring biomolecular dynamics and mechanics [50]. Among various AuNP-assisted imaging techniques, the special optical properties of AuNPs are commonly used. Plasmon-enhanced fluorescence (PEF) phenomena, the ability of gold NPs to increase the excitation of fluorescence, have been used to obtain imaging [51]. Indeed, Masuda et al. showed that this method is useful in achieving a high-resolution image of the interaction between nanomaterials and fluorescence-labeled actin filaments in basophilic rat leukemia cells [52]. It is worth mentioning that small gold nanoparticles, known as quantum dots (QDs), can display intrinsic long life-time fluorescence and negligible photobleaching [30]. However, due to their small size, they cannot support the SPR effect, but they exhibit fluorescence in the visible or near-infrared spectrum (NIR) [53,54]. Zhang et al. manufactured nanoprobes based on gold nanoclusters, showed their cellular uptake, and displayed in vivo images of tumor sites in mice. The biodistribution and excretion pathway studies of the nanoprobes in tumor-bearing nude mice revealed their superior penetration and retention behavior in tumors [55]. 

Additionally, the unique properties of gold nanoparticles allow the detection of specific components of complex biological mixtures. For this purpose, Raman spectroscopy is used. The Raman effect is a by-product of photon scattering interacting with matter, leading to the obtaining of a chemical fingerprint of a molecule. AuNPs greatly enhance this effect. This phenomenon is called surface-enhanced Raman scattering (SERS) and can be applied to highly sensitive probes [50,56,57]. In particular, non-spherical, irregular AuNPs, which exhibit a much stronger electromagnetic field, have been investigated for use in the SERS imaging method [58]. It is a particularly desirable method due to the possibility of establishing the exact distributions of neoplastic cells, the multifocality of which may cause minor recurrence or metastasis. Harmsen et al. presented a method of visualizing tumor margins using a new generation of surface-enhanced resonance Raman scattering nanostars, with a gold core, in the form of a nanoprobe. They obtained a picture of the full extent of tumors in mouse models of breast cancer, sarcoma, pancreatic ductal adenocarcinoma, and prostate cancer [56]. Sánchez-Purrà et al. designed a multiplex assay using the SERS method in combination with a lateral flow assay (LFA), which differentiates the biomarkers of Zika virus protein from non-structural protein 1 (NS1) of dengue virus. Gold nanostars were conjugated with specific antibodies for both virus-induced diseases, and 15-fold and 7-fold lower limits of detection (LOD) were obtained compared to colorimetric tests for ZIKV NS 1 and DENV NS 1, respectively. This method not only improved the diagnosis of diseases but also helped in the early diagnosis of human risk when biomarker levels are still low [59]. 

Diagnostic approaches based on nanotechnology have also proved to be a promising alternative to conventional and low-cost effective methods of detecting miRNAs in body fluids or tissue samples [60]. MicroRNAs, or small regulatory RNAs, can be used as biomarkers in the early diagnosis of cancer. However, they occur in low concentrations and are difficult to detect [61]. Li et al. developed a colorimetric miRNA detection method based on the isothermal exponential amplification reaction (EXPAR)—AuNP-assisted amplification. EXPAR provides a linear detection range of 50 fM to 10 nM miRNA with a detection limit of 46 fM in 60 min. The method also detects single-nucleotide differences between homologous nucleic acids [62]. Amal et al. verified the possibility of diagnosing stomach cancer based on a breath test. The volatile organic compounds that appear in exhaled breath can be linked to disease condition. The samples from patients were analyzed using the gas chromatography linked to mass spectrometry (GCMS) method, and a nanoarray sensor composed of gold nanoparticles and single-wall carbon nanotubes (SWCNTs) covered with different ligands was used. The obtained results indicate that nanoarrays could be a tool for screening gastric cancer and related precancerous lesions, as well as for monitoring the effectiveness of therapy [63].

The main problem of commonly used contrast agents in magnetic resonance imaging, such as gadolinium-based contrast agents (GBCA), is the possible release of free heavy metals in vivo [64]. In 2006, gadolinium (Gd) was linked to a debilitating and potentially fatal condition called nephrogenic systemic fibrosis (NSF), a rare disease of unknown cause that affects patients with renal insufficiency treated with Gd as a contrast agent [65]. It is also possible that Gd accumulates in tissues over a long period of time [66]. The brain is particularly at risk of accumulation of contrast agents. Due to the less developed lymphatic system, it is difficult to wash it out. Although the clinical consequences of the accumulation of contrast agents are unclear, the search for new compounds with improved magnetic properties is ongoing [67]. The solution may lie in the use of gold nanoparticles. The absence of contrast agent accumulation in the lungs, liver, and spleen of mice was demonstrated by Alric et al. through the use of gadolinium chelate—functionalized gold nanoparticles [68]. 

AuNPs can serve, not only as carriers, but they can also be combined with superparamagnetic materials, such as iron oxides, e.g., Fe_3_O_4_ [69]. Iron oxide nanoparticles are the only NP-based contrast agents that have been approved by the Food and Drug Administration (FDA) [70]. They are an example of negative contrast agents, which means that they make the enhanced parts of a T2-weighted image darker, while shortening the T2 relaxation time [71]. Fe_3_O_4_@AuNPs combine, not only unique magnetic properties, but also their gold coating endows the material with photothermal, photodynamic, or SERS capabilities and allows functionalization with antibodies, ensuring targeted action. Iancu et al. investigated the physical and biological properties of nanoparticles of this type and obtained stable nanoparticles that produced a negative T2 signal in vivo when injected into rats, which means that they can be used as negative contrast in MRI. A reduction in cytotoxicity was also observed [67]. In 2019, research was carried out on the use of gold nanoparticles in the diagnosis of breast cancer. Chauhan et al. used GBCA-functionalized spherical gold NPs and a cancer-targeted DNA aptamer for MRI imaging. They showed increased contrast agent uptake with relevant breast cancer cell lines compared to non-targeted control counterparts [72]. Currently, however, the use of contrast agents with gold NPs is limited to imaging small animals due to autofluorescence, strong light absorption and scattering, which significantly reduces spatial resolution [68]. 

One of the obstacles in using optical methods is that the penetration depth for tissue samples is limited to several hundred µm, mostly due to strong scattering within the tissues [50]. Photoacoustic imaging (PAI) is a relatively new imaging method that is rapidly gaining ground in the context of biomedical imaging applications. It is a hybrid method that uses the same properties as fluorescence imaging and also combines ultrasound detection. Compared to other methods, PAI imaging allows to obtain images of deep tissues, up to 7 cm. A nanosecond pulsed laser is used here; the molecules absorb optical energy and convert it into heat, inducing a temperature change, generating acoustic waves that are detected by ultrasonic transducers [73,74,75]. Gold nanoparticles have been considered as excellent photoacoustic contrast agents, mainly due to their large absorption cross section tuned to the optical window (700–1000 nm), which minimizes endogenous absorption and maximizes imaging depth [76]. Copland et al. demonstrated the use of bioconjugated gold nanoparticles as a tool for deep imaging of breast cancer cells. The sensitivity of this assay showed that concentrations low as 109 NPs per milliliter were detectable at a depth of 6 cm [77]. 

The use of gold nanoparticles in the treatment of cancer depends on their size, and thus the ability to penetrate tissues. Size also affects nanoparticle accumulation, systemic toxicity, and half-life [78]. It has been shown that, the so-called ultra-small gold nanoparticles, i.e., <10 nm in size, in particular, have a high tissue penetration capacity, lower toxicity, and better renal cleansing efficiency [79,80,81]. Chen et al. used ultra-small gold nanoparticles as a contrast agent for PET (positron emission tomography) imaging. They obtained copper-labelled gold nanoparticles which exhibited efficient and rapid renal clearance from the kidneys. Thus, gold nanoparticles can be potentially used for the diagnosis of kidney disease [82].

Gold nanoparticles, in addition to being used as a diagnostic tool in the oncological environment, can also be used in ophthalmology. Many ophthalmic imaging methods are available, but are not sufficient to diagnose eye disease at the molecular level before morphological changes become visible. In particular, the optical coherence tomography (OCT) method, using gold nanoparticles, has attracted a lot of attention [83]. OCT is a widely used, non-destructive, and non-invasive technique that provides real-time images of tissue sections with a deep resolution. The limitation of this technique is the inability to provide information about the physiology or molecular processes of the examined tissue, which can be achieved using an exogenous contrast agent with a high optical scattering coefficient [84]. The AuNP plasmon band can be modulated by changing the diameter or modification to the surface using different ligands, so that they can readily absorb light at the wavelengths used in OCT [85]. The AuNP-deposited biosensing paper strip created by Kim et al. allows for early detection of infectious keratitis or conjunctivitis. The strip was made using the successive ionic layer absorption and reaction (SILAR) technique, which helps in maximizing SERS activity of surface plasmons. By analyzing human tears, it was possible to distinguish between healthy and infected eyes, as well as to determine the type of infection [86].

Genetics could also benefit from the use of gold nanoparticles, especially the detection of single nucleotide polymorphisms. Currently, the liquid biopsy method, which is used to detect DNA samples in patient’s blood, is associated with the most modern laboratory methods and professional personnel are required to perform these analyses. Nanomaterials offer an opportunity to create easy-to-use diagnostic devices that operate on the colorimetric principle, without the need to perform advanced techniques, which could speed up cancer detection [87]. Chengnan et al. proposed a novel and simple optical biosensor for DNA detection based on unmodified gold nanowires as signal and hybridization chain reaction (HCR) transducers. They were able to detect target DNA in the range of 0–60 nM, with a detection limit of 1.47 nM. This biosensor exhibits a wide linear range, high sensitivity, and selectivity of DNA detection [88].

## 3. Therapy

Tunable surface chemistry of gold nanoparticles, as well as their unique properties, facilitate their coating, functionalization, and integration with a host of biomolecular moieties opens the door to a wide gamut of applications in therapy [89]. Among various approaches, photothermal therapy (PTT), where gold NPs are applied, is known as a less invasive technique in cancer treatment [90]. It is based on the conversion of light energy (usually in the NIR region) into heat to induce subsequent cellular necrosis or apoptosis (Figure 4) and causes tumor ablation [91]. Unfortunately, healthy tissues are also exposed to damage during this process; therefore, methods with improved specificity and precise spatial–temporal selectivity of PTT are currently under investigation [92]. To achieve these features, AuNPs for PTT could be functionalized with specific tumor-targeting molecules [90]. Passive and active targeting are commonly used methods, in which poly (ethylene) glycol or selected molecule/antibody specific to tumor markers are introduced [93]. In fact, the FDA approved PEGylated gold NPs as the most successful NP, and are being used in ongoing human pilot studies [94]. PEGylated nanoparticles accumulate preferentially in tumor tissues due to the enhanced permeability and retention effect (EPR) exhibited by solid tumor [95]. Hirsch et al. achieved irreversible photothermal ablation of tumor tissue in mice after exposure of PEGylated gold nanoshells to NIR light [96]. In addition, PEGylation of nanoparticles saves them from immunorecognition and prolongs their blood circulation [94,97]. The accumulation of intratumoral PEGylated particles (after 2, 6, and 24 h) was checked in solid HSC-3 solid tumor in mice after intravenously injection via the tail vein [98]. High particle loading was observed following 24 h of circulation. These authors also compared the efficiency of PTT therapy with direct and intravenously administered AuNPs. The results reported dramatic tumor size decreases in mice either for directly-injected (resorption of ˃57%) or intravenously-treated (25%) tumors with AuNPs after NIR laser irradiation. The selectivity of the PTT treatment may be increased by conjugating NPs to antibodies or other disease-specific molecules. The efficiency of gold nanospheres as a photothermal agent was demonstrated by selective delivery of AuNPs to oral squamous carcinoma cells that overexpress EGFR, a clinically related cancer biomarker [99]. Another molecule used for PTT in cancer treatment is folic acid (FA), the tumor-targeting ligand, due to its high binding affinity to folate receptors [100]. These receptors are often overexpressed on the surface of tumor cells, thus, combining them with AuNPs may be useful in PTT [101]. Zhang et al. reported a high efficacy of targeted delivery of FA-attached nanomaterials to melanoma cells. They also showed that the photothermal killing of cancer cells depends on the temperature in PTT. An alternative promising approach to cancer treatment is photodynamic therapy (PDT), in which the activation of photosensitizer (PS) occurs in response to exogenously applied light, which in turn triggers the generation of reactive oxygen species (ROS) and cause cell death (Figure 4) [102]. However, due to the low solubility of PS in the physiological environment [94], and their limited delivery due to insufficient depth of penetration [103], a new strategy, the PDT/PTT dual approach, is being investigated [104]. Photosensitizers can be conjugated to AuNPs (nanorods, nanostars, nanoclusters, etc.) through different approaches based on self-assembly [105]. In this dual therapy, light-induced heating can be exploited to either induce heating to release a chemical payload or to generate reactive oxygen species to induce cell death [94]. Wang et al. covalently anchored Chlorin e6 (Ce6, a commonly used photosensitizer) on the surface of gold nanostars (GNS) to perform simultaneous PDT/PTT treatments of breast cancer and lung cancer models at different irradiation times, both in vitro and in vivo [106]. The authors proved the theranostic potential of using this nano-device. The heat-inducing effect after being applied to gold nanostructures was also obtained for radiofrequency therapy (RF) when electromagnetic waves at radio frequencies were introduced [92]. Cardinal et al. reported non-invasive radiofrequency technique coupled with gold nanoparticle to the thermal ablation of tissue and cancer cells, in both in vitro and in vivo systems [107]. PTT and/or PDT therapies may also be applicable as a therapeutic strategy for infections caused by bacteria or fungi. Zharov et al. have showed a new laser nanoparticle-based PTT for antimicrobial therapy [108]. By using combined techniques, AuNPs and laser, a successful killing rate of *Staphylococcus aureus* was achieved. Various modification of AuNPs and their conjugates with other molecules have been used for the selective destruction of microorganism by light exposure [104].

Gold nanoparticles have also found application in cancer immunotherapy. This type of therapy works by stimulating the natural immune system to attack cancer cells (Figure 5) [109]. However, immunomodulators used for this purpose have several disadvantages, including high cost, instability, limited half-life, and rapid drug clearance [110]. Systemic toxic reactions are an additional problem in immuno-oncological treatment [111]. The safety and efficacy of immunotherapy can be improved by modifying the developed biomaterials, including various therapeutic agents, increasing the targeting of specific immune responses [112]. AuNPs can, not only provide an ideal platform for loading immunomodulators, but are also biocompatible, which allows them to be administered intravenously, and thus have the potential to accumulate in tumor cells, which is especially useful for vaccines or adjuvants [113]. A vaccine consisting of AuNPs functionalized with the MUC-1 protein developed by Mocan et al. acted as a strong macrophage activator, resulting in the release of cytokines in murine peritoneal macrophages, and thus demonstrates a strong antitumor effect [114].

AuNS are one of the most promising platforms due to their remarkable geometry, which greatly enhances light absorption and provides a high photon-to-heat conversion efficiency due to the plasmonic effect [110]. Liu et al. created a new nanoplatform by preparing GNS@CaCO_3_/Ce_6_ nanoparticles attached to natural killer cells (NK cells). They applied it in a lung cancer model, demonstrating a significant inhibitory effect on the growth of A549 cells in vitro and in vivo [115]. 

AuNPs facilitate delivery to the immune system, promote the therapeutic effect of antigens and adjuvants, and have an adjuvant effect themselves. This has been demonstrated in studies of the delivery of ovalbumin OVA and CpG adjuvant using gold nanoparticles in an in vivo B16-OVA tumor model. The use of AuNP, not only increased the efficacy of both agents and induced strong antigen-specific responses, but, also, the delivery of AuNP-OVA alone promotes significant antigen-specific responses. Therefore, AuNPs are effective carriers of peptide vaccines with the potential to use lower and safer doses of adjuvant during vaccination [116].

AuNS has also been used in a new treatment of glioblastoma. The SYMPHONY therapy (synergistic immunophotothermal nanotherapy) combines treatments using gold nanostars as photothermal inducers and laser-induced photothermal therapy with checkpoint blockade immunotherapy. Mice cured using this therapy successfully rejected reintroduction of cancer cells, which means that, thanks to the SYMPHONY treatment, they gained anticancer memory [117]. Liang et al. combined PPT/PDT therapy using gold nanostars fabricated with HER-2 monoclonal antibody and near-infrared region (NIR) photosensitizer indocyanine green (ICG). AuNS@ICG-Ab was loaded on CIK cells (Cytokine-induced killer cells). The nanoplatform was able to migrate into SK-BR-3 tumor cells, accumulate efficiently, activate the immune response, and, through the specific link between trastuzumab and cells, the platform could enhance the effect of PDT/PTT therapy, leading to inhibiting the progression of tumors in mice models [118].

## 4. Delivery System

The tunable size, shape, and functionality of the gold nanoparticles make them great carriers for delivering other compounds (Figure 6). To assess whether nanoparticle-based technology can be successfully used in the clinic, it is important to understand the biodistribution of nanoparticles, which primarily affects their effectiveness and safety [11]. To be able to assess the effectiveness of the passive or active therapeutic targeting of a nanoparticle, it is important, first of all, to understand the extent to which nanoparticles locate at the appropriate site. The potential unintended accumulation in other tissues should also be monitored and the long-term fate of nanoparticles should be characterized to understand the safety of the administered nanoparticles [2,11]. Various parameters influence the cytotoxicity of gold nanoparticles; the main factors to be taken into account are the size, type of cell, penetration capacity, as well as distribution and absorption in tissues [7]. A drug delivery system (DDS) provides many advantages to the distribution of chemicals within the body, mainly because of improvements to their stability, solubility, and biodistribution [6]. Enhanced drug accumulation with using nanoparticle-based DDS may occurs by passive (utilizing EPR effect) or active (biofunctionalization with molecules) targeting [92]. Especially, compared to traditional approaches for delivering chemotherapeutic drugs, which cause side effects on healthy tissues, AuNP-targeting delivery is a promising strategy for diseases treatment. Through proper functionalization, the particles can locate specific tissues. Since they can be easily taken up by the cell, they can also accumulate in specific subcellular compartments [119]. The presence of a negative charge on the surface of gold nanoparticles makes them easily modifiable. Thus, they can be functionalized by the attachment of monoclonal antibodies, drugs, peptides, and genes [94]. 

Paciotti et al. studied gold nanoparticles as a drug delivery vector for tumor necrosis factor alpha (TNFα); AuNPs were functionalized with polyethylene glycol (PEG) and tumor necrosis factor (TNF) [120]. Colloidal gold conjugated with PEG and TNF showed a specific accumulation in mouse colon tumors (MC-38) to a greater extent than free TNF, and was more effective in reducing tumor volume at a lower dose than native TNF, and also exhibited minimal systemic toxicity [121]. In Aurimune (CytImmune), a PEG linker was used to attach recombinant human TNF to gold nanoparticles (Figure 7). A phase 1 study showed that patients with advanced cancer tolerate Aurimune well. It was also found that drug accumulation in tumor masses through the EPR effect was aided by the PEG layer, which reduces uptake by the mononuclear phagocyte system. Gold-based nanodrugs have not yet been approved by the FDA [122].

Tumor tissue generally has a leaky neovessels, which allow nanoparticles to penetrate and accumulate easily [123]. Methotrexate (MTX), a drug for cancer treatment, upon conjugation with AuNPs, exhibited a faster rate of accumulation and higher cytotoxicity towards various tumor cell lines compared to free MTX [124]. Biodistribution and proper release of the drug tethered to the NPs is dependent on the type of the linker used to their functionalization. Wang et al. developed a system with doxorubicine (DOX) attached to the surface of AuNPs with acid-labile linkages [125]. When it was introduced to MCF-7/ADR cancer cells, highly efficient cellular entry, intracellular release of DOX, and enhanced retention were observed when compare either to free DOX or DOX attached using carbamate linkages. Hormone receptors of the tumor can be targeted using AuNP drug delivery systems. In the study by El-Sayed and colleagues, they used tamoxifen to both target and deliver AuNP to breast cancer cells that overexpress estrogen receptor [126]. By targeting the estrogen receptor, tamoxifen (TAM) was chemically modified using PEG-thiol and covalently bound to 25 nm AuNPs with approximately 12,000 TAMs per particle. AuNP-TAM conjugates demonstrated a 2.7-fold greater efficacy in human breast cancer cells (MCF-7) compared to native TAM due to increased intracellular transport of tamoxifen by nanoparticle endocytosis [120,127]. El-Sayed and colleagues developed another drug delivery system targeting elevated levels of androgen receptors in prostate cancer. Androgen receptor targeting was achieved by modifying antiandrogen chemotherapeutic agents with PEG-thiol to facilitate binding to 30 nm AuNPs. There was an approximately 10-fold increase in the therapeutic efficacy of antiandrogens in chemotherapy-resistant prostate cancer cells (DU-145), after modification and binding of the antiandrogen drug molecule to AuNP [128,129]. Gold NPs can shift unfavorable pharmacokinetics of biocomponents. Some authors have reported an increase in the half-time of peptide–drug conjugates (PDCs) from minutes (administered alone) to hours (upon AuNPs conjugation) without influence on the cytotoxicity [130]. Combination of passive and active targeting delivery, could increase the effectiveness of therapy by enhanced tumor selective accumulation and controlled release of loaded drugs in response to stimuli [92]. Moreover, multi-component conjugates should reduce the concentration of chemotherapeutic agent required for therapeutic usefulness, while simultaneously reducing systemic toxicity [12]. Gold nanoparticles have also been utilized as carriers for efficient DNA/RNA delivery in gene therapy [123]. Motoi et al. reported that a construct of the PEGylated AuNPs functionalized with SH-siRNA is able induce gene knockdown in the human hepatic cancer cell line and exhibited prolonged serum half-life [131]. By means of AuNP, and under the influence of ionizing radiation, the DNA-damaging effect of platinum-based anticancer drugs, such as oxaliplatin, carboplatin, and cisplatin can be enhanced. Using the enhanced carbodiimide/N-hydroxysuccinimide coupling reaction used to internalize multiple platinum sites in several cancer cell lines, multivalent gold oligonucleotide nanoparticles (DNA-S-AuNP) terminally linked to a cisplatin prodrug (platinum complex [IV]) were developed. The reductive release of cisplatin from the prodrug was caused by a decrease in intracellular pH and this caused intra-strand 1,2-d (GpG) cross-links in the cell nucleus, as confirmed by an antibody specific for the adduct, with cytotoxicity in some cases exceeding that of cisplatin [127,132]. Another study demonstrated that gold nanoparticles chemically functionalized with alkylthiol-terminated oligonucleotides were highly stable and could bind complementary nucleic acids [133]. Thus, these properties have made oligonucleotide-functionalized AuNPs the center piece of several highly sensitive and selective assays for biomolecule detection [134]. For example, Nam et al. developed a nanoparticle-based bio-barcode for the detection of prostate-specific antigen (PSA) which may be useful in biosensing applications [135]. Gold nanoparticles are also promising tool for delivery of a cancer vaccine, because they preferentially accumulate within tissues and cells of the immune system [123]. Lee et al. functionalized gold NPs with red fluorescence protein (RFP, as a model cancer antigen) and with a CpG oligonucleotide (a short single-stranded synthetic DNA used as vaccine adjuvant) [136]. Following this, the nanotool was injected to melanoma tumor-bearing mice. After interaction of this antigen with dendritic cells, antitumor activity expressed by inducing antibody production was observed. AuNPs have promise to be used as drug carriers to increase the therapeutic efficacy of monoclonal antibodies. Studies that have been of interest to many researchers in recent years have focused on the interaction of cetuximab-conjugated AuNP (cetuximab-AuNP) with tumors overexpressing EGFR. It has been shown that cetuximab-AuNP showed increased EGFR endocytosis and inhibition of downstream signaling, leading to inhibition of cell proliferation and acceleration of apoptosis compared to AuNP or cetuximab alone. The use of cetuximab-AuNPs may be a non-standard chemotherapy and radiotherapy approach as increased uptake by specific targeting followed by improvement of therapeutic agents/radiation may be a new procedure for the treatment of EGFR overexpressing tumors [137]. Gold nanoparticles have been used as a valuable component in cosmeceutical industries due to their antimicrobial activity and in beauty care due to the ability to accelerate blood circulation, as well as anti-inflammatory and antiseptic properties or by causing delay to the aging process [138].

## 5. Conclusions

Systems based on nanoparticles offer interesting opportunities in various fields of science and technology, including biomedicine. Gold nanoparticles have been tested in various clinical trials (Table 1), but their translation into practice has been slow. The main obstacles are methods of their synthesis that are not sterile, repeatable, and scalable, and their behavior in biological systems is not fully understood [139,140]. However, recent advances in nanomaterial synthesis and production methods have made it possible to control factors such as the size, morphology, composition, and surface chemistry using precise technology, which allows obtaining nanostructures that are more stable in biological systems [7].

Nanoparticle-based technologies for biomedical applications are still being developed for in vivo applications. The physicochemical properties of nanoparticles have been well researched, but the same cannot be said for their biological properties, so there is a continuing need to understand the biodistribution and mechanism of the potential removal of nanostructures. Two main reasons why gold nanoparticles are among the most widespread and are a good choice in many studies are their chemical inertness and minimal toxicity, meaning that they can pass through the body without causing adverse reactions. Studies show that AuNPs coated with polyethylene glycol (PEG) reduce cell toxicity, which allows them to be used intravenously as carriers of DNA and drugs. PEG is amphiphilic in nature, which enables the stabilization and monodispersity of gold nanoparticles in an aqueous biological environment. PEG also increases the circulatory half-life of AuNPs, allowing uptake and removal of nanoparticles by the reticuloendothelial system [7,141]. 

In conclusion, because of the minimal toxicity of gold and the ease of controlled synthesis and functionalization of AuNPs, it has become interesting to use AuNPs in therapy, diagnosis, imaging, and drug delivery. Despite these benefits, AuNPs must undergo necessary safety studies before they can be used in a clinical setting.

## Figures and Tables

**Figure 1 materials-15-03355-f001:**
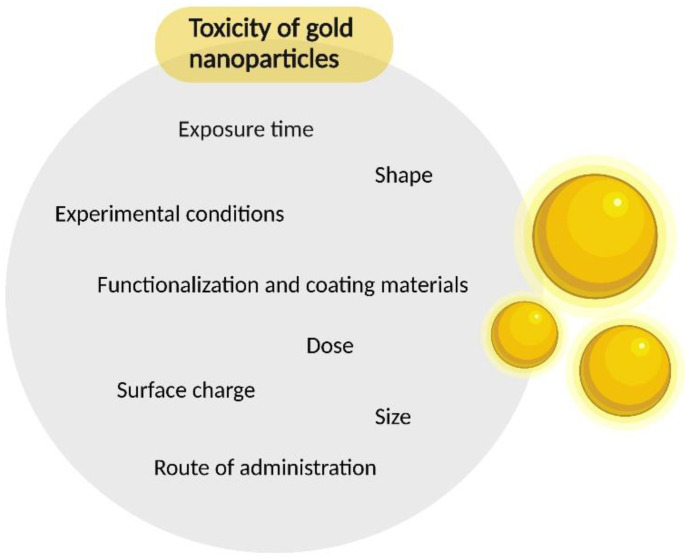
Factors influencing the toxicity of gold nanoparticles.

**Figure 2 materials-15-03355-f002:**
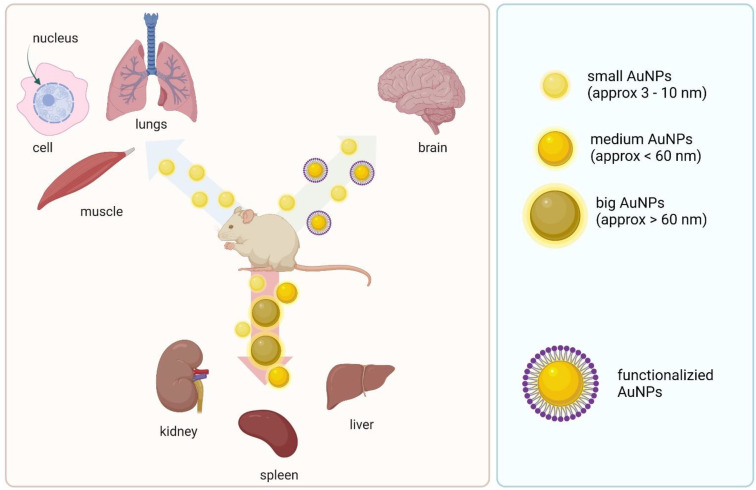
Biodistribution of different size gold nanoparticles.

**Figure 3 materials-15-03355-f003:**
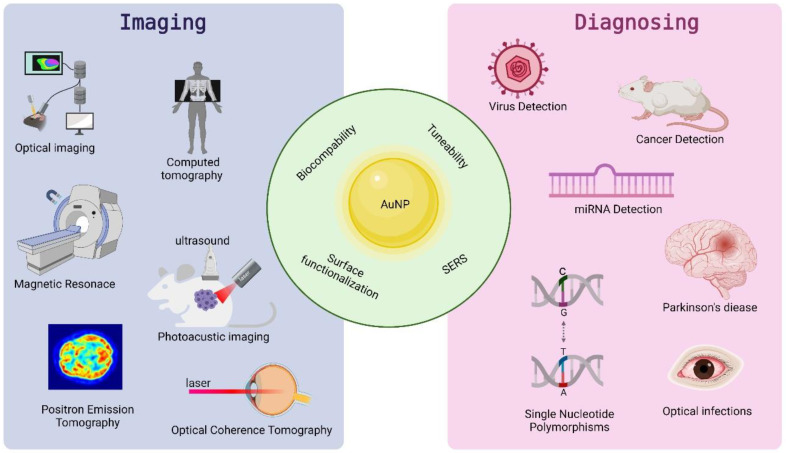
Properties and application of AuNPs in imaging and diagnosing.

**Figure 4 materials-15-03355-f004:**
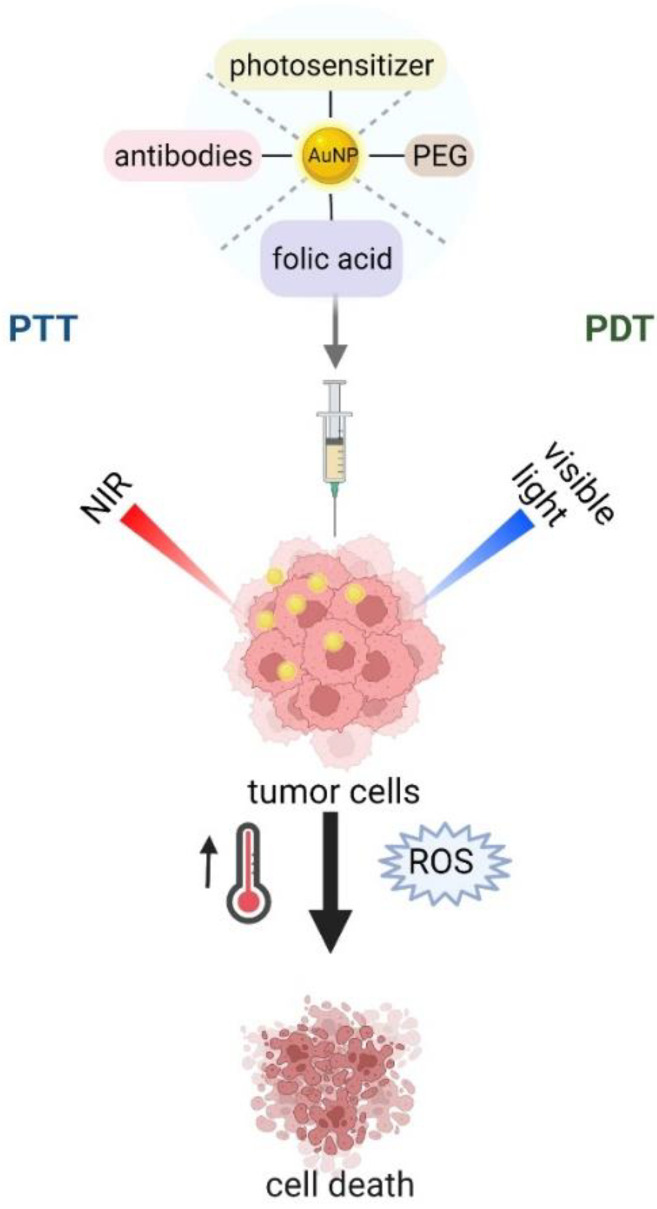
Schematic illustration of phototherapy.

**Figure 5 materials-15-03355-f005:**
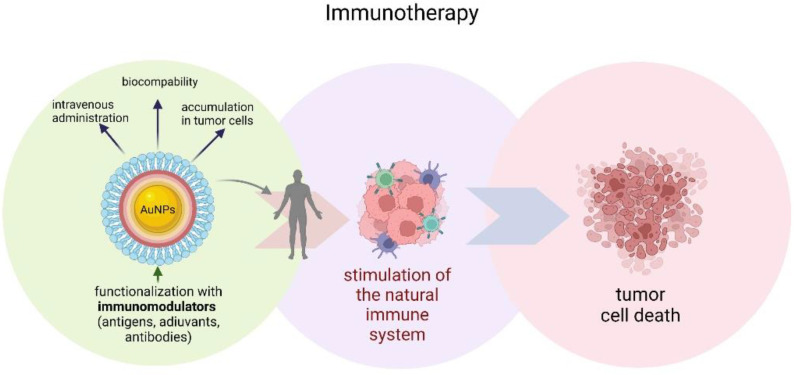
Schematic illustration of gold nanoparticles immunotherapy.

**Figure 6 materials-15-03355-f006:**
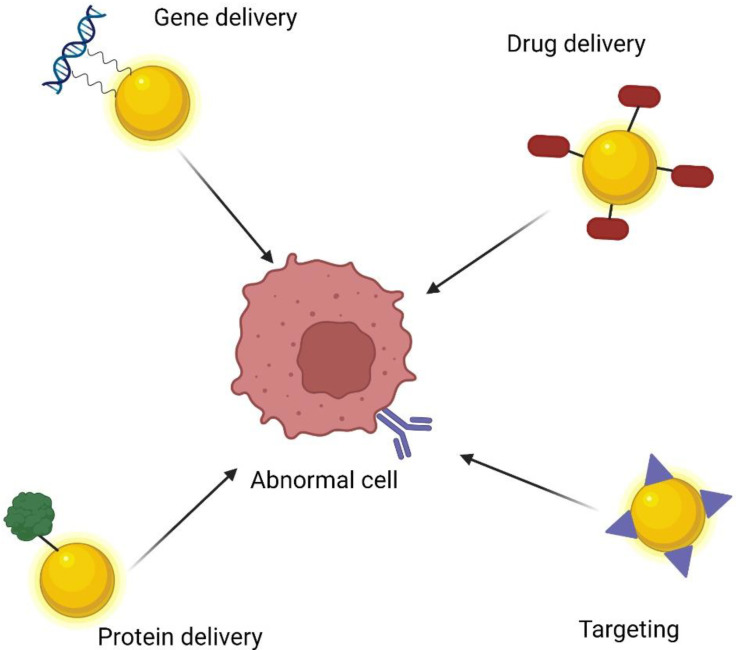
Gold nanoparticles as carriers of various compounds.

**Figure 7 materials-15-03355-f007:**
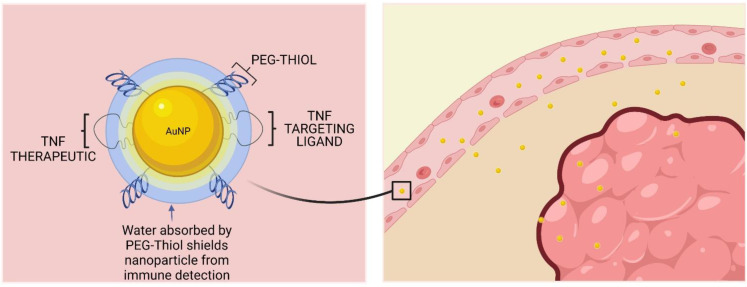
Design and mode of action of CYT-6091 (Aurimune^®^). The blood vessels of tumors are weak, resulting in gaps in the walls. When CYT-6091 reaches these blood vessels, the nanoparticles are small enough to pass through the resulting cavities to their target, the tumor.

**Table 1 materials-15-03355-t001:** Gold nanoparticles clinical trials for imaging and diagnosis.

Materials	Application	Clinical trials.gov Identifier
Gold nanoparticles	Sensors functionalized with gold nanoparticles. Organic functionalized gold nanoparticles. Detection of gastric lesions.	NCT01420588
Gold nanoparticles	Exhaled breath olfactory signature of pulmonary arterial hypertension.	NCT02782026
Functionalized carbon nanotubes and gold nanoparticle	Exploratory study using nanotechnology to detect biomarkers of Parkinson’s disease from exhaled breath.	NCT01246336
CD24-gold nanocomposite	Diagnostic and prognostic accuracy of gold nanoparticles in salivary gland tumors.	NCT04907422

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
