# Peer review of "Treasure on the Earth—Gold Nanoparticles and Their Biomedical Applications"

_materials, 2022, doi:10.3390/ma15093355_

Round 1

Reviewer 1 Report

The manuscript assessed “Treasure on the earth – gold nanoparticles and their biomedical applications”, this research is under the scope of this Materials Journal, the topic is relevant for readers and this research deals with potentially significant knowledge to the field.

However, there are Major concerns about the present manuscript : 

(Abstrat)  

  • It is important to show more information about this review.

(Keywords)  

  • Please order the keywords alphabetically for a standardized presentation of the keywords

(Introduction)

What is the importance of this review study? What is the gap on this field of literature?

    • You do not think this study are include to the others already done? Which results are comparable with others study? What has this study new?
    • Many  Antimicrobial peptides (AMPs), such as LL37 peptides, may be immobilized nanoparticules (AuNPs) on the surface of medical devices, to render them with antimicrobial and angiogenic properties. Comune 2021 https://doi.org/10.1039/D1BM01034D; Akilesh Rai, 2022  https://doi.org/10.1039/D1TB02617H
  • The Introduction, it seems too short. Authors should be focus on the area for the readers.
  • Improve the resolution quality of all figures and graphs (and a presentation). The font/ language in the figure/caption is different from the text.
  • Authors must say where they carried out the bibliographic research, which databases were consulted. On the other hand, the exclusion/inclusion criteria are not found in the text, it would be pertinent if they had used this methodology. This section would better communicate to readers if restructured. A flowchart or diagram of the article selection would be valuable.

(M&M) 

  • You need to explain the process of articles selection, using a PRISMA flow diagram.  We need to know what was the reasons for exclusions or inclusion. 

(Conclusions)

Despite the article being extensive, the conclusions should be shorter and more objective.

(Author contributions) 

Please please write the contributions of each author.

(References)

  • Need to add more reference in manuscript, see the articles recommend.
  • Check reference’s format MDPI in the manuscript, and in the references. The titles of references have a different format, the title of the article is written in capital letters at the beginning of words, others only in lower case. Also, the standardized format of presentation in the journal's name. Because names have written in a different format, one is not abbreviated, others are not.

Reviewer 2 Report

In this review article, the authors described the biomedical applicability of Au nanostructures. The manuscript appears to be interesting and can be reconsidered for publication after a revision. 

1) Page 2, lines 82-84 should be deleted (template instructions). 
2) At least a brief chapter discussing the Au toxicity in-vivo should be introduced. 
3) Au is non-digestible material and the main issue is to remove it from the body after treatment. In this case, size plays an important role and deserves a better description in this review.  
4) CT imaging capability should be described quantitatively, i.e. in terms of Hounsfield units, following works can be useful for example DOI: 10.1038/srep26177 and DOI: 10.1016/j.nanoso.2021.100712.
5) In my humble opinion, the authors can add several additional graphs/schemes to support the text description.  

Round 2

Reviewer 1 Report

This research is under the scope of this journal; the topic is interesting for readers.

The authors improved the quality of the manuscript after the reviewer's indications.

Reviewer 2 Report

A revised manuscript can be accepted for publication.